# TEST-TIME ADVERSARIAL DEFENSE WITH OPPOSITE ADVERSARIAL PATH AND HIGH ATTACK TIME COST

## ABSTRACT

Deep learning models are known to be vulnerable to adversarial attacks by injecting sophisticated designed perturbations to input data. Training-time defenses still exhibit a significant performance gap between natural accuracy and robust accuracy. In this paper, we investigate a new test-time adversarial defense method via diffusion-based recovery along opposite adversarial paths (OAPs). We present a purifier that can be plugged into a pre-trained model to resist adversarial attacks. Different from prior arts, the key idea is excessive denoising or purification by integrating the opposite adversarial direction with reverse diffusion to push the input image further toward the opposite adversarial direction. For the first time, we also exemplify the pitfall of conducting AutoAttack (Rand) for diffusion-based defense methods. Through the lens of time complexity, we examine the trade-off between the effectiveness of adaptive attack and its computation complexity against our defense. Experimental evaluation along with time cost analysis verifies the effectiveness of the proposed method.

## 1 INTRODUCTION

### 1.1 BACKGROUND

It has been well known that deep learning models are vulnerable to adversarial attacks by injecting (imperceptible) adversarial perturbations into the data that will be input to a neural network (NN) model to change its normal predictions (Athalye et al. (2018); Carlini et al. (2019); Croce et al. (2023); Frosio & Kautz (2023); Goodfellow et al. (2015); Gowal et al. (2021); Madry et al. (2018); Venkatesh et al. (2023)). Please also see Chen & Liu (2023) for a recent review on the adversarial robustness of deep learning models. It can be found from the literature that adversarial attacks defeat their defense counterparts easily and rapidly, and there is still a gap between natural accuracy and robust accuracy.

The study of adversarial defense in resisting adversarial attacks can be divided into two categories: (1) Adversarial training/Training-time defense (Gowal et al. (2021); Hsiung et al. (2023); Huang et al. (2023); Suzuki et al. (2023); Wang et al. (2019; 2023); Wu et al. (2020); Zhang et al. (2019)); and (2) Input pre-processing/Test-time defense (Alfarra et al. (2022); Chen et al. (2022); Hill et al. (2020); Ho & Vasconcelos (2022); Nie et al. (2022); Wang et al. (2022); Wu et al. (2022); Yoon et al. (2021)). Adversarial training utilizes adversarial examples derived from the training data to enhance the robustness of the classifier. Despite the effort in training-time defense, we do see ( RobustBench Croce et al. (2021)) there is still a remarkable gap between natural accuracy and robust accuracy.

Different from the training-time defense paradigm, in this paper, we propose a new test-time adversarial defensive method by pre-processing data in a way different from the prior works. It is a kind of purifier and serves as a plug-and-play module that can be used to improve the robustness of a defense method once our module is incorporated as a pre-processor. Specifically, the formulation of processing the input data is derived as: $\min_{\phi, \theta} \mathbb{E} \left[ \max_{x' \in B(x)} \mathcal{L}((f_\phi \circ g_\theta)(x'), y) \right]$, where $x'$ denotes the adversarial example corresponding to clean image $x$ with label $y$, $B(\cdot)$ is the threat model, $f_\phi$ is the image classifier parameterized by $\phi$, and $g_\theta$ is a pre-processor.

A key to test-time defense is the design of pre-processor or denoiser (*e.g.*, $g_\theta$), which aims at denoising an adversarial example to remove the added perturbations. Intuitively, the goal is to have the denoised image as close to the original one so as to achieve perceptual similarity.

## 1.2 RELATED WORKS

We introduce representative test-time adversarial defense methods (Alfarra et al. (2022); Ho & Vasconcelos (2022); Hill et al. (2020); Yoon et al. (2021); Nie et al. (2022); Wang et al. (2022); Wu et al. (2022)) that share the same theme as our method. Please also see Sec. 6 in the Supplementary for details of Hill et al. (2020); Yoon et al. (2021); Wang et al. (2022); Wu et al. (2022).

In Alfarra et al. (2022), a defense method is proposed by connecting an anti-adversary layer with a pre-trained classifier $f_\phi$. Given an input image $x$, it will be first sent to the anti-adversary layer for generating anti-adversarial perturbation $\gamma$ by solving an optimization problem. As the name implies, in most cases, the direction $\gamma$ will be opposite to the direction of adversarial perturbation. The resultant purified image $x + \gamma$ is then used for classification.

DISCO (Ho & Vasconcelos (2022)) is proposed as a purification method to remove adversarial perturbations by localized manifold projections. The author implemented it with an encoder and a local implicit module, which is leveraged by the architecture called LIIF (Chen et al. (2021); Chen & Zhang (2019)), where the former produces per-pixel features and the latter uses the features in the neighborhood of query pixel for predicting the clean RGB value.

In DiffPure (Nie et al. (2022)), given an input (clean or adversarially noisy), the goal is to obtain a relatively cleaner version through a series of forward and reverse diffusion processes. Moreover, a theoretical guarantee is derived that, under an amount of Gaussian noise added in the forward process, the adversarial perturbation may be removed effectively. This is independent of the types of adversarial perturbations, making DiffPure defend against unseen attacks.

Recently, the robustness of diffusion-based purifiers is considered overestimated. Lee & Kim (2023) provides recommendations for robust evaluation, called *surrogate process*, and shows that the defense methods may be defeated under the surrogate process. Kang et al. (2024) proposes DiffAttack, a new attack technique against diffusion-based adversarial purification defenses, that can overcome the challenges of attacking diffusion models, including vanishing/exploding gradients, high memory costs, and large randomness. The use of a segment-wise algorithm allows attacking with much longer diffusion lengths than previous methods.

Although the aforementioned purification-based adversarial defense methods show promising performance in resisting adversarial attacks, Croce et al. (2022) argues that their evaluations are ineffective in two aspects: (i) Incorrect use of attacks or (ii) Attacks used for evaluation are not strong enough. However, the authors also mentioned test-time defense complicates robustness evaluation because of its complexity and computational cost, which impose even more computations for the attackers.

## 1.3 MOTIVATION

Let us take image classification as an example, where clean/natural accuracy is the classification accuracy for benign images and robust accuracy is measured for adversarial samples. However, we argue that "perceptually similar" does not mean adversarial robustness as it is not guaranteed to entirely remove the adversarial perturbations such that the residual perturbations still have an impact on changing the prediction of a learning model. On the contrary, we propose to purify the input data along the direction of opposite adversarial paths (OAPs) excessively, as illustrated in Fig. 1.

Conceptually, if we add the adversarial perturbation along the opposite direction of Projected Gradient Descent (PGD) (Madry et al. (2017)), denoted as "$-adv$," to a given data, robust accuracy can be improved. To gain an insight that excessive denoising (more than one step along the opposite gradient) is advantageous in resisting attacks, a simple experiment was conducted by moving each data point $x$ to the new position $x^K$ through $K$ iterations of opposite adversarial perturbation, according to the ground truth label and classifier. Given each kind of $x^K$, the accuracy change is illustrated in Table 1. For the decrease in clean accuracy at $K = 1$, we conjecture that the process "$-adv$" is still unstable. Hence, some data points near the decision boundary may be perturbed to incorrect class. Moreover,

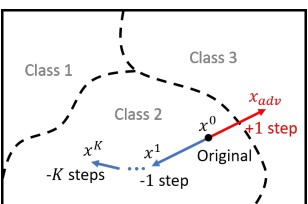

| -$K$ Steps | Clean Accuracy (%) | Robust Accuracy (%) |
|---|---|---|
| 0 | 95.16 | 0.18 |
| -1 | 89.94 | 6.18 |
| -3 | 98.56 | 83.64 |
| -5 | 99.63 | 96.78 |
| -10 | 99.96 | 99.86 |
| -20 | 100.00 | 100.00 |

Figure 1: Concept diagram of new reference point generation via $K$ consecutive purifications along opposite adversarial paths (OAPs).

Table 1: Pre-processing the training dataset by adding $K$ steps of $-adv$ (via PGD (Madry et al. (2017)); see Fig. 1) and feeding to a non-defense classifier (ResNet-18 (He et al. (2016))) pre-trained on CIFAR-10 (Krizhevsky et al. (2009)) for testing. $K = 0$ indicates original data.

motivated by Croce et al. (2022), our defense method also aims to complicate the computation of adaptive adversarial attacks.

### 1.4 CONTRIBUTIONS

Different from prior works, the concept of OAP can be incorporated into any training scheme of purifiers, and the OAP-based purifier can also become a part of modules in other defense processes. For instance, OAP-based purifiers can provide additional directions within reverse diffusion, whereas diffusion models alone (Song et al. (2020) (baseline model in DiffPure)) only provide direction to generate images. Unlike the traditional purification methods, we do not use *classifier-generated labels* (*e.g.* Anti-Adv Alfarra et al. (2022)) in our baseline purifier during testing. On the contrary, combining the proposed baseline purifier with the reverse diffusion process provides reference directions pointing to a safer area during the purification process.

Contributions of this work are summarized as follows:

1. We are first to present the idea of excessive denoising along the opposite adversarial path (OAP) as the baseline purifier for adversarial robustness. (Sec. 3.1)

2. We integrate the OAP baseline purifier and conditional reverse diffusion as a sophisticated adversarial defense that can be interpreted as moving purified data toward the combination of directions from the score-based diffusion model and baseline purifier (Sec. 3.2).

3. To complicate the entire defense mechanism by complicating the computation overhead of adaptive attacks accordingly, we study a double diffusion path cleaning-based purifier (Sec. 3.3). This creates a trade-off between the attack effectiveness and attack computation.

4. For the first time, we exemplify the pitfall of conducting AutoAttack (Rand) for diffusion-based adversarial defense methods (Sec. 3.4).

## 2 PRELIMINARY

### 2.1 BASIC NOTATION

In the paper, $x$ denotes an input image, $\hat{x}$ denotes a recovered image or overly denoised/purified image, $x_{adv}$ denotes an adversarial image, $y$ is a ground-truth label of $x$, $\hat{y}$ is a prediction, $g_\theta$ is a purifier, and $f_\phi$ is a pre-trained classifier.

For the diffusion model, the forward process is denoted by $q(\cdot|\cdot)$ and the backward/reverse process is denoted by $p_\theta(\cdot|\cdot)$ with parameter $\theta$. For $t \in [0, T]$, $x_t$ represents an image at time step $t$ during the forward / reverse diffusion process. Usually, $x_0$ is a clean image and $x_T \sim \mathcal{N}(0, I)$.

For the adversarial attack, it modifies the input image $x$ by adding to it adversarial perturbation $\delta$ by calculating the gradient of loss according to information leakage of pre-trained NN $f_\phi$ without changing $\phi$, causing $f_\phi$ to classify incorrectly. According to the leakage level, there are roughly two types of attacks. Please see Sec. 7 in the Supplementary for details.

## 2.2 Diffusion Models

Since the diffusion model (Sohl-Dickstein et al. (2015); Ho et al. (2020); Song & Ermon (2019); Song et al. (2020)) is a baseline model in diffusion-based purifiers, to make this paper self-contained, please refer to Sec. 8 of Supplementary for a brief introduction to the diffusion model.

## 3 Proposed Method

We describe the proposed test-time adversarial defense method with its flowchart illustrated in Fig. 2.

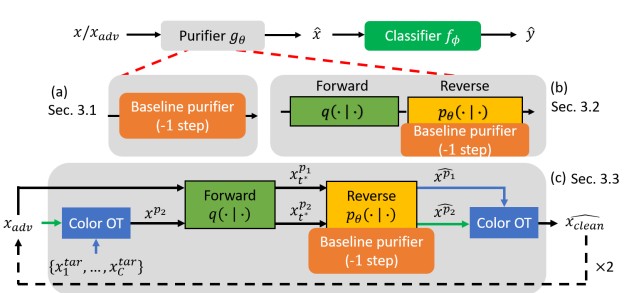

Figure 2: Flowchart of our method. The purifier (gray block) can be one of (a)-(c), where (a) is the proposed baseline purifier, (b) shows the combination of baseline purifier and reverse diffusion, and (c) expands (b) with two diffusion paths. In (c), $x_1^{tar}, \ldots, x_C^{tar}$ are obtained via Eq. (2) from fixed $C$ images with one image per class (in CIFAR-10). The image in front of Color OT with green/blue arrow is called the source/target image. $x^{p_2}$ is defined in Eq. (10).

### 3.1 Baseline Purifier: Opposite Adversarial Path (OAP)

Given a classifier model $f_\phi$ parameterized by $\phi$, a loss function $\mathcal{L}(x, y, \phi)$, and a pair of data $(x, y)$, the adversarial attack can be computed as

$$x_{adv} = \Pi_{x+\mathcal{S}}(x + \alpha \operatorname{sign}(\nabla_x \mathcal{L}(x, y, \phi))), \tag{1}$$

where $\mathcal{S}$ is the set that allows the perceptual similarity between natural and adversarial images. This iterative process aims to find the adversarial image $x_{adv}$ that maximizes the loss function.

On the other hand, the opposite direction of each iteration points to minimize the loss. Assume now we get an ordinary noisy input $x_{adv} = x + \delta$ with $\|\delta\|_p \leq \epsilon_p$ via image processing, a denoiser can push the denoised input close to $x$ within a non-perceptual distortion. Nevertheless, if the noisy input $x_{adv}$ is a sophisticated design via adversarial attack, it is too early to claim, depending on the perceptual similarity between $x$ and $x_{adv}$, that the denoised image can be free from being affected by adversarial perturbations. We argue that if we properly push the denoised image further away from the decision boundary, the downstream classifier can still successfully classify the input since the direction we push points to a lower loss area on the input-loss surface, as illustrated in Fig. 1. Also note that the *plug-and-play* module lies under the setting that the baseline purifier is only trained on a given attack (*e.g.*, PGD-$\ell_\infty$-7), which is independent of the attacks (*e.g.*, PGD-$\ell_\infty$-40, AutoAttack, and BPDA+EOT) used in testing. In addition, the diffusion model is pre-trained (Sec. 3.2) and does not involve adversarial examples during its training.

#### 3.1.1 New Reference Point Generation

Previous test-time defense methods with a plug-and-play fashion take $x_{adv}$ as an input and generate the predicted "clean" image $\hat{x}$. In our scenario, we want to move a few steps further. Starting from the clean image $x$, ground-truth label $y$, parameter $\phi$ and loss function $\mathcal{L}$ of classifier $f$, we can generate a new reference point $x^K$ *for training* by the following formula:

$$x^k = \Pi_{x^{k-1}+\mathcal{S}}(x^{k-1} - \alpha \operatorname{sign}(\nabla_{x^{k-1}} \mathcal{L}(x^{k-1}, y, \phi))), \tag{2}$$

for $1 \leq k \leq K$, where $x^0 = x$. If we iterate Eq. (2), we can get a series of data, $x^1, x^2, \ldots, x^K$, as illustrated in Fig. 1.

| -K steps | Non-adapt PGD-$\ell_\infty$ / ResNet-18 | | Non-adapt AA / WRN-28-10 | | BPDA / VGG16 | |
|---|---|---|---|---|---|---|
| | Clean Acc (%) | Robust Acc (%) | Clean Acc (%) | Robust Acc (%) | Clean Acc (%) | Robust Acc (%) |
| 0 | 89.57 | 73.13 | 89.00 | 85.00 | 88.38 | 47.37 |
| -1 | 90.71 | 86.10 | 91.66 | 88.79 | 89.26 | 56.34 |
| -3 | 89.53 | 82.02 | 90.23 | 86.14 | 87.78 | 59.94 |
| -7 | 89.33 | 56.21 | 89.58 | 69.04 | 88.42 | 52.31 |

Table 2: Evaluation of DISCO trained with the relation between new reference point and adversarial perturbations by PGD attack generated in ResNet-18. Entire CIFAR-10 testing dataset was used. (Left) Attack: Non-adaptive PGD-$\ell_\infty$. Test model: ResNet-18. (Middle) Attack: Non-adaptive AutoAttack (AA). Test model: WRN-28-10 (Zagoruyko & Komodakis (2017)). (Right) Attack: BPDA. Test model: VGG16 (Simonyan & Zisserman (2015)).

### 3.1.2 Baseline Purifier Training

In traditional denoising, the goal is to train a purifier that produces a denoised output $\hat{x}$ from the adversarial input $x_{adv}$, denoted as $x_{adv} \mapsto \hat{x}$, such that $\hat{x}$ and $x$ can be as similar as possible in terms of, say, $\ell_p$-norm. We, instead, train the purifier to produce $\widehat{x^K}$ from $x_{adv}$ that further points toward the opposite adversarial attack direction. We call the resultant $\widehat{x^K}$ an excessively-denoised image and the model $g_\theta$ that moves data along the opposite adversarial path (OAP) the "baseline purifier."

In practice, we train a baseline purifier using data pairs $\{(x_{adv}, x^K)\}$ with a certain number of opposite steps $K \in \mathbb{N}$, where $x^K$ is generated by Eq. (2). The training procedure of $g_\theta$ is to minimize:

$$\theta^* = \underset{\theta}{\arg\min} \left\| g_\theta(x_{adv}) - x^K \right\|_1, \tag{3}$$

where $g_\theta$ can be any existing defense methods (*e.g.*, DISCO Ho & Vasconcelos (2022)). The results of training on different opposite steps are shown in Table 2 respect to PGD-$\ell_\infty$ (Madry et al. (2017)), AutoAttack (AA) (Croce & Hein (2020)), and BPDA (Athalye et al. (2018)). We can observe that the idea of the new reference point indeed improves DISCO. Specifically, when $K = 1$, the robust accuracy can be improved greatly, but it decreases as $K$ goes larger. The results are somewhat inconsistent with those in Table 1. The reason we conjecture is that the experiment presented in Table 1 is under the condition of using the ground-truth label to move data step-by-step, whereas that in Table 2 is not. Hence, as $K$ increases excessively, the distance that pushes the data increases excessively, which is similar to the effect of large step size in gradient descent. Therefore, based on the empirical observations, we will empirically set $K = 1$ for learning the opposite direction of an adversarial attack during training.

On the other hand, we will later demonstrate that OAP is a powerful module readily to be incorporated with existing adversarial defenses (*e.g.*, DISCO) in improving both the clean and robust accuracy.

### 3.2 Diffusion-based Purifier with OAP Prior

In Sec. 3.1, we have witnessed the merit of baseline purifier based on OAP in improving robustness against adversarial attacks. This data moving trick also motivates us to study how to incorporate OAP prior and diffusion models as a stronger adversarial defense.

We first propose to integrate the idea of opposite adversarial paths with the reverse diffusion process (*e.g.*, guided diffusion Dhariwal & Nichol (2021), ILVR Choi et al. (2021), and DDA Gao et al. (2023)) to achieve a similar goal of pushing the input image further toward the opposite adversarial direction. More importantly, for each step in the reverse diffusion process, the purifier is used to provide a direction that points to $x^K$.

To this end, according to Eq. (14) of guided diffusion described in Sec. 8 in Supplementary, by taking logarithm and gradient with respect to $x_{t-1}$ (Dhariwal & Nichol (2021)), we can derive

$$\nabla_{x_{t-1}} \log p_\theta(x_{t-1}|x_t, y) = \nabla_{x_{t-1}} \log p_\varphi(x_{t-1}|x_t) + \nabla_{x_{t-1}} \log p_\phi(y|x_{t-1}), \tag{4}$$

where $t$ denotes the diffusion time step. Based on Langevin dynamics, we get a sampling chain on $x_{t-1}$ as:

$$x_{t-1} \leftarrow x_t + \nabla_{x_{t-1}} \log p_\varphi(x_{t-1}|x_t), \tag{5}$$

where we get the first direction (specified by $p_\varphi$) of moving to $x_{t-1}$. However, if we want to generate $x_{t-1}$ by moving along the direction given $x_t$ and $y$, we have to introduce the second direction (specified by $p_\phi$) to move to $x_{t-1}$ given condition $y$ based on Eq. (4). Hence, we add $\nabla_{x_{t-1}} \log p_\phi(y|x_{t-1})$ in the sampling chain (5) as:

$$x_{t-1} \leftarrow x_t + \nabla_{x_{t-1}} \log p_\varphi(x_{t-1}|x_t) + \nabla_{x_{t-1}} \log p_\phi(y|x_{t-1}), \tag{6}$$

where the last two terms are the same as the RHS of Eq. (4). Note that the second term can be approximated by a model $\epsilon_\varphi(\cdot)$ that predicts the noise added to the input. According to (11) in Dhariwal & Nichol (2021), it can be used to derive a score function as:

$$\nabla_{x_{t-1}} \log p_\varphi(x_{t-1}|x_t) = -\frac{\epsilon_\varphi(x_{t-1})}{\sqrt{1 - \bar{\alpha}_t}}, \tag{7}$$

where $\bar{\alpha}_t = \prod_{s=1}^{t}(1 - \beta_s)$.

Different from previous works, if $y$ in the third term of Eq. (6) is replaced with the new reference point $x^K$, as described in Eq. (2) of Sec. 3.1, then the term becomes $\nabla_{x_{t-1}} \log p_\phi(x^K|x_{t-1})$ and represents how to move along the direction to $x^K$ given $x_{t-1}$. This can be set by

$$\widehat{x^K} \leftarrow g_\theta(x_{t-1}); \quad \nabla_{x_{t-1}} \log p_\phi(x^K|x_{t-1}) \approx \eta(\widehat{x^K} - x_{t-1}), \tag{8}$$

where $\eta$ is the step size and $g_\theta(\cdot)$ is the purifier (see Sec. 3.1) that can approximate the mapping of $x_{adv} \rightarrow x^K$. Hence, the purification process can be interpreted as moving toward the combination of directions from the score-based diffusion model (Nie et al. (2022); Song & Ermon (2019); Song et al. (2020)) and baseline purifier $g_\theta(\cdot)$.

### 3.2.1 CONNECTING THE OAP PRIOR WITH DIFFUSION

We are aware that the base purifier has to operate in the domain the same as that in the diffusion reverse process, *i.e.*, they deal with different inputs with noises at different scales. However, according to Eq. (3), the baseline purifier only takes inputs that are adversarially perturbed. Hence, during the training of baseline purifier, we randomly add different scales of noise to the input data so that the base purifier can accommodate the different noise scales in the reverse diffusion process, denoted as:

$$\theta_n^* = \arg\min_\theta \mathbb{E}_{p_{data}(x_{adv})} \mathbb{E}_{p_{\sigma_t}(\tilde{x}|x_{adv})} \left\| g_\theta(\tilde{x}) - x^K \right\|_1, \tag{9}$$

where $t$ is uniformly chosen from $0 \dots t^*$, $\sigma_t$ is the corresponding noise scale at diffusion time step $t$, and $\tilde{x}$ is the perturbed data according to the diffusion process. We replace the baseline purifier $g_\theta$ in Eq. (8) in Sec. 3.2 with this purifier $g_{\theta_n}$.

### 3.3 DIFFUSION PATH CLEANING-BASED PURIFIER

In this section, we describe how to further utilize other gradients from different constraints to modify/move our samples toward specific directions. Moreover, the goal is to complicate the entire framework of purifier+classifier so as to complicate the computation of adaptive attacks accordingly while maintaining comparable clean and robust accuracy. We first conduct a test to verify whether such a framework could be affected by such an attack.

In this test, we verify the framework composed of two diffusion paths, denoted as $p_1$ and $p_2$, and a pre-trained classifier $f_\phi$ (*e.g.*, pre-trained WRN-28-10), as shown in Fig. 3. The adaptive adversarial image $x_{adv}$ is generated via BPDA+EOT (Athalye et al. (2018)) as an input to path $p_1$ while the clean image $x$ is assumed to be available (ideal case) in path $p_2$. In this case, we minimize the $\ell_2$ distance between the intermediate image in the reverse process $p_1$ and that in $p_2$, which gives a direction to make $p_1$ close to $p_2$. Finally, the output $\widehat{x^{p_1}}$ is feed into the classifier $f_\phi$ for prediction. We obtain the natural accuracy of $93.5\%$ and robust accuracy of $93.0\%$ from the test. This provides us a hint that the diffusion path $p_1$ should be maintained relatively clean (*e.g.*, both the input and reverse diffusion process in path $p_1$ are as clean as those in path $p_2$) so that the output of $p_1$, which is the recovered image $\widehat{x^{p_1}}$, is purified enough.

Therefore, the motivation here is to expand the idea of the opposite adversarial direction in modifying (i) the input for arriving at a safer area and (ii) the entire path for purification. Nevertheless, the clean

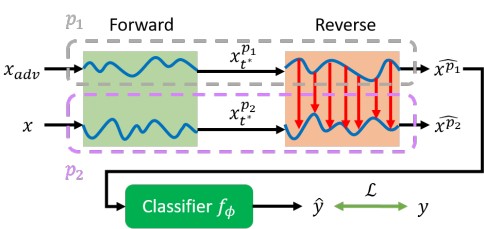

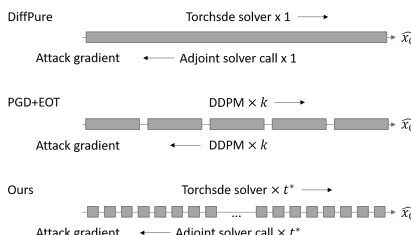

Figure 3: (Ideal model) Red arrows depict directions to minimize $\ell_2$ distance between the intermediate images of two reverse diffusion paths, $p_1$ and $p_2$. $\mathcal{L}$ is the loss function. Dataset: CIFAR-10.

Figure 4: Reverse diffusion process implementations: The original implementation of DiffPure involves only one function call in reverse and adjoint solver calls. The PGD+EOT attack utilizes a surrogate diffusion process with fewer steps than purification steps. However, in our implementation, we use the same number of steps for purification and attack.

image $x$ corresponding to $x_{adv}$ required for the second path $p_2$ is absent during testing. In addition, it is known that adversarial perturbation is added to an image and causes imperceptible changes. In view of this, we resort to generating purified images as input to $p_2$ using the new reference point strategy, as described in Eq. (2) of Sec. 3.1.

Conceptually, the idea of generating the input to path $p_2$ that guides path $p_1$ is to transfer pixel values from the source image (adversarial image) to the other target image (clean/purified image), which can be treated as finding the optimal transport plan that moves every 3D point (RGB value) in a source point cloud to a target point cloud with the minimum cost (*e.g.*, in terms of $\ell_2$ distance between two point clouds). Fortunately, we can use non-attack images, which are the training data, combined with Eq. (2) to produce excessively denoised target images for diluting the attack perturbation.

Based on the above test and observations, we now describe the proposed method for cleaning the diffusion path with adversarial images as input. The flowchart is illustrated in Fig. 2(c). First, suppose we have $x_{adv}$ as the source image, it will be processed by color transfer with optimal transport (Feydy et al. (2019)), which is denoted as "Color OT" in Fig. 2(c), using the images coming from the training dataset. To this end, we pick $C$ images with one image per class, where $C$ stands for the number of classes. By using Eq. (2) to generate new reference points from these picked images, we have the target images $x_1^{tar}, \ldots, x_C^{tar}$ for "Color OT" to change/purify the adversarial pixels in $x_{adv}$. The $C$ target images will not be picked again throughout the testing so that there is no randomness.

Second, after finding the $x_j^{tar}$ that has the lowest Sinkhorn divergences $S_\varepsilon$ (Eq. (3) in Feydy et al. (2019)) with $x_{adv}$, we then use color transfer $f_{CT}$ to modify $x_{adv}$ with reference to $x_j^{tar}$. The output is denoted as $x^{p_2}$. The purification procedure is specified as:

$$j = \underset{i \in \{1,\ldots,C\}}{\operatorname{argmin}} S_\varepsilon(x_{adv}, x_i^{tar}); \quad x^{p_2} = f_{CT}(x_{adv}, x_j^{tar}). \tag{10}$$

As our starting point, $x^{p_2}$ goes into the diffusion process, as shown in Fig. 2(c). This ensures all pixel values in $x^{p_2}$ are not from $x_{adv}$. To make it clear, examples of the intermediate images generated from the diffusion process in Fig. 2(c) are illustrated in Fig. 5 of Sec. 9 in the Supplementary.

Third, we put $x_{adv}$ and $x^{p_2}$ into the diffusion model and set $t^*$, which is the optimal time step (Nie et al. (2022)) to remove the adversarial noise. We maintain two paths: the path with superscript $p_1$ for denoising the color values and the path with superscript $p_2$ for recovering the image. Unlike the test in Fig. 3, during the reverse diffusion process, we do not use $p_2$ to pull $p_1$, since $x^{p_2}$ is generated by $f_{CT}$. Instead, we use "baseline purifier+reverse diffusion" described in Sec. 3.2 on one path, $p_1$. Therefore, after the reverse diffusion process, the image $\widehat{x^{p_2}}$ will refer to the denoised image $\widehat{x^{p_1}}$ as the target for $f_{CT}$ to restore the colors, which is denoted as $\widehat{x_{clean}}$. This is because $\widehat{x^{p_2}}$ is still a color-transferred image after the diffusion model, but the output from $p_2$ in the aforementioned test (Fig. 3) starts from the ideal clean image $x$ without needing color restoration. The whole process will be iterated again with starting point $\widehat{x_{clean}}$ and $t^*$ being halved at each iteration. Please see Algorithm 1 in Sec. 12 of Supplementary in describing the entire procedure.

### 3.4 GRANULARITY OF GRADIENT APPROXIMATION IN REALIZING POWERFUL ADAPTIVE AUTOATTACK

We present to implement a more powerful adaptive AutoAttack via granularity of gradient approximation in order not to overestimate robustness. Actually, our implementation requires the output in each step from `torchsde` in the diffusion reverse process, which starts from $x_{t^*}$ and calls `torchsde` to produce the output $x_{t^*-1}$ of next time step till we get the final image $\hat{x}$. Hence, if one understands the mechanism of using *adjoint method* as BPDA correctly, gradient computation in the reverse diffusion process will demand the same amount of calls of *adjoint method* as in that of `torchsde`. We have to particularly point out that this is different from DiffPure (Nie et al. (2022)), where the authors only used one `torchsde` call for the final image and one call of *adjoint method* for computing the gradient. We believe the granularity (one call vs. multiple calls of *adjoint method*) of gradient approximation causes the performance difference, and the use of multiple calls indeed provides AutoAttack with sufficient information to generate a more powerful adversarial perturbation.

To verify our finding, we have observations across different datasets, as shown in Table 3. First, we selected a subset from CIFAR-10 testing dataset consisting of $64$ images, then generated the corresponding adversarial examples from adaptive AutoAttack (Rand) with 20 EOT via two different implementations, including (1) AutoAttack (Rand-DiffPure): Original code from DiffPure (Nie et al. (2022)) using one `torchsde` function call and (2) AutoAttack (Rand-Ours): Our own implementation that pulls the output $x_t$ at each time step from `torchsde` solver, which means 100 `torchsde` function calls. See Fig. 4 for comparison of different implementations of reverse diffusion process.

We can see from Table 3 that in comparison with AutoAttack (Rand-DiffPure), the defense capability of DiffPure is remarkably reduced (the accuracy in boldface) when the adversarial examples generated from AutoAttack (Rand-Ours) are present, obviously indicating robustness overestimation. Actually, it is evidence of revealing that our implementation can let attackers create stronger adversarial examples and can be used as a proxy to attack diffusion-based purifiers. Also, this finding sheds light on whether using *adjoint method* hides the information used for creating stronger adversarial examples in an adaptive AutoAttack setting.

Finally, since DiffPure (Nie et al. (2022)) has not been evaluated in Croce et al. (2022), it is believed that this simple trick of implementation that creates stronger AutoAttack (Rand) can be an easy way of attacking test-time adversarial defense purifiers and a promising supplement to Croce et al. (2022). In the following experimental evaluations, this kind of adjoint strategy will be used in implementing stronger adaptive attacks.

| AutoAttack | (Rand-DiffPure) | (Rand-Ours) | AutoAttack | (Rand-DiffPure) | (Rand-Ours) | AutoAttack | (Rand-DiffPure) | (Rand-Ours) |
|---|---|---|---|---|---|---|---|---|
| DiffPure | 76.56% | **64.06%** | DiffPure | 26.56% | **20.31%** | DiffPure | 46.88% | **28.13%** |

Table 3: Robust accuracy for adversarial examples (Adv) generated from different implementations of diffusion purification under adaptive AutoAttack (Rand) with 20 EOT. Our implementation uses output in every time step from `torchsde`, whereas DiffPure (Nie et al. (2022)) uses `torchsde` without accessing the intermediate outputs, which is encapsulated in `torchsde` function call. Left: CIFAR-10/WRN-28-10; Middle: CIFAR-100/WRN-28-10; Right: ImageNet/ResNet-18.

### 3.5 ATTACK COST AND TIME COMPLEXITY

We study how to resist adaptive attacks by analyzing and increasing the time cost of breaking the proposed defense models. The results are shown in Table 5. Due to space constraints, please see the time complexity analysis in Sec. 10 in the Supplementary for details.

## 4 EXPERIMENTS

We examine the performance of proposed test-time adversarial defense methods, described in Sec. 3.2 and Sec. 3.3, against state-of-the-art adversarial attacks, and performance comparison with SOTA purification-based defenses.

### 4.1 DATASETS AND EXPERIMENTAL SETTINGS

Three datasets, CIFAR-10 (Krizhevsky et al. (2009)), CIFAR-100 (Krizhevsky et al. (2009)), and ImageNet (Deng et al. (2009)), were adopted, where the results for CIFAR-100 and ImageNet are shown in Table 3 and Sec. 11 of Supplementary. All experiments were conducted on a server with Intel Xeon(R) Platinum 8280 CPU and NVIDIA V100.

For a fair comparison, we followed RobustBench (Croce et al. (2021)) and existing literature to conduct experiments on two popular NN models, including ResNet-18 (He et al. (2016)) and WRN-28-10 (Zagoruyko & Komodakis (2017)). The step size, $\eta$, in Eq. (8) of Sec. 3.2 was set as $2.5 \times 10^{-3}$ and we followed Nie et al. (2022) to set $t^*$ used in Sec. 3.2 and Sec. 3.3 as 0.1. Since $t^* = 0.1$, the number of steps required in the reverse process is 100, where the step size $dt$ for `torchsde` solver is set to 1e-3. We set $\varepsilon = 0.05$ in Eq. (10), which is the default setting in the official package (GeomLoss) (Feydy et al. (2019)). For all attacks, we used $\ell_\infty$ and set perturbation $\|\delta\|_\infty \leq 8/255$.

For training, the only model that needs to be trained is the baseline purifier $g_{\theta_n}$ with $K = 1$, which we chose DISCO (Ho & Vasconcelos (2022)) as the baseline to be combined with our new reference point generation in Eq. (2) with $K = 1$ throughout the experiments. In computing the attack gradient per step ($K$), we used PGD-$\ell_\infty$ with 7 iterations. For testing the diffusion-based purifiers, we followed the testing paradigm described in DiffPure (Nie et al. (2022)), including the uses of 24 random subsets (each contains 64 images) for AutoAttack and 15 random subsets (each contains 200 images) for BPDA+EOT from CIFAR-10 testing dataset.

### 4.2 ADVERSARIAL ROBUSTNESS EVALUATIONS

Two types of adversarial attacks, including non-adaptive attack (PGD-$\ell_\infty$ Madry et al. (2017), AutoAttack (Standard) Croce & Hein (2020)) and adaptive attack (BPDA+EOT Athalye et al. (2018), PGD+EOT Lee & Kim (2023) and DiffAttack Kang et al. (2024)), were adopted. For AutoAttack, we utilized the package AutoAttack (Croce & Hein (2020)) with $\ell_\infty$, in which it has two settings: (1) "Standard," which includes APGD-CE, APGD-DLR, FAB, and Square Attack and (2) "Rand," which includes APGD-CE and APGD-DLR with Expectation Over Time (EOT) (Athalye et al. (2018)) in case of models with stochastic components. To the most extreme case in which the attacker knows every detail about our framework of "purifier+classifier," we utilized BPDA (*adjoint method* Li et al. (2020) for the diffusion model) to bypass purifiers and EOT to combat the randomness in purifiers. As mentioned in Sec. 3.4, our adjoint strategy will be used to implement stronger adaptive attacks in order to avoid robustness overestimation.

The robustness performance was measured regarding clean/natural accuracy (Clean Acc) for benign samples and robust accuracy (Robust Acc) for adversarial samples. Several test-time adversarial defense methods, including Anti-Adv (Alfarra et al. (2022)), DISCO (Ho & Vasconcelos (2022)), DiffPure (Nie et al. (2022)), SOAP (Shi et al. (2021)), Hill *et al.* (Hill et al. (2020)), and ADP (Yoon et al. (2021)), were adopted for comparison. Tables 4 and 5 show the robustness evaluation results and indicate that our methods either outperform or are comparable with the prior works.

The experiment in Table 4 is under the setting of non-adaptive attacks (PGD-$\ell_\infty$ with 40 iterations and AutoAttack (Standard)), in which the attacker only knows the information of the downstream classifier. According to Alfarra et al. (2022), we specifically point out that the authors used the robustly trained classifier, Adversarial Weight Perturbation (AWP) (Wu et al. (2020)), as the testing classifier. So, except Alfarra et al. (2022), we used a normally trained classifier throughout the experiments.

Table 5 shows the results obtained under adaptive attacks, including stronger ones like PGD+EOT (Lee & Kim (2023)) and DiffAttack (Kang et al. (2024)). For the two kinds of AutoAttack (Rand) described in Sec. 3.4, please refer to Table 3. Since most diffusion-based purifiers exhibit randomness, we utilized the "EOT" setting for randomness, and "BPDA" for bypassing the reverse process of diffusion-based methods, which use the *adjoint method* to calculate the gradient of such process. We also provide the time needed to attack an image (attack time cost) against a defense method. Besides, according to the dual-paths design of Sec. 3.3, all adaptive attacks have to attack both paths. As a result, our defense experiences TWICE stronger attacks than other single-path methods since gradients are obtained from two paths. In other words, attacks are computed twice.

| Defense Methods | Clean Accuracy (%) | Robust Accuracy (%) | Attacks |
|---|---|---|---|
| No defense | 94.78 | 0 | PGD-$\ell_\infty$ |
| AWP (Wu et al. (2020))* | 88.25 | 60.05 | AutoAttack (Standard) |
| Anti-Adv (Alfarra et al. (2022))* + AWP (Wu et al. (2020)) | 88.25 | 79.21 | AutoAttack (Standard) |
| DISCO (Ho & Vasconcelos (2022))* | 89.26 | 82.99 | PGD-$\ell_\infty$ |
| DISCO (Ho & Vasconcelos (2022)) **+ our OAP** ($K=1$) | 92.5±2.06 | 88.29±3.3 | PGD-$\ell_\infty$ |
| DiffPure (Nie et al. (2022)) | 88.06±2.65 | 87.21±2.28 | PGD-$\ell_\infty$ |
| DiffPure (Nie et al. (2022)) | 88.15±2.86 | 87.71±2.12 | AutoAttack (Standard) |
| SOAP (Shi et al. (2021))* | 96.93 | 63.10 | PGD-$\ell_\infty$ |
| Hill *et al.* (Hill et al. (2020))* | 84.12 | 78.91 | PGD-$\ell_\infty$ |
| ADP ($\sigma = 0.1$) (Yoon et al. (2021))* | 93.09 | 85.45 | PGD-$\ell_\infty$ |
| Ours (Sec. 3.2) | 90.77±2.25 | **88.48±2.04** | PGD-$\ell_\infty$ |
| Ours (Sec. 3.2) | **90.46±2.36** | **89.06±2.62** | AutoAttack (Standard) |

Table 4: Non-adaptive robustness comparison between our method and state-of-the-art methods. Classifier: WRN-28-10. Asterisk (*) indicates that the results were excerpted from the papers. Boldface indicates the best performance for each attack. Note that, by incorporating our *Opposite Adversarial Path* (OAP) prior, the clean and robust accuracy of DISCO can be greatly increased.

| Defense Methods | Clean Accuracy (%) | Robust Accuracy (%) | Attack time cost (sec.) | Attacks |
|---|---|---|---|---|
| No defense | 94.78 | 0 | N/A | BPDA+EOT |
| DiffPure | **92.38±1.86** | 80.92±3.53 | 592.92 | BPDA+EOT |
| Hill *et al.*.* | 84.12 | 54.90 | N/A | BPDA+EOT |
| ADP ($\sigma = 0.1$)* | 86.14 | 70.01 | N/A | BPDA+EOT |
| Ours (Sec. 3.3) | 92.08±1.99 | **81.25±3.62** | **6880.97** | BPDA+EOT |
| DiffPure | 96.88 | 46.88 | 3632.94 | PGD+EOT |
| Ours (Sec. 3.3) | **100** | **53.12** | 22721.90 | PGD+EOT |
| DiffPure | 89.02 | 46.88 | N/A | DiffAttack |
| Ours (Sec. 3.3) | **95.31** | **93.75** | 20397.27 | DiffAttack |

Table 5: Aadaptive robustness comparison between our method and state-of-the-art methods with attack time cost per image. Classifier: WRN-28-10. Asterisk (*) indicates that the results were excerpted from the papers. Boldface indicates the best performance for each attack. The attacks include BPDA+EOT, PGD+EOT (Lee & Kim (2023)), and DiffAttack (Kang et al. (2024)).

More specific, we can see from Table 5 that, in addition to accuracy, the time costs the attackers need to generate attack examples for our defense method are greatly higher than those for other defense methods. If the attackers would like to shorten computations of generating adversarial examples, the number of iterations of conducting attacks or the number of EOT need to be reduced, thereby weakening the attack performance. Take BPDA+EOT as an example: the total time to finish a batch testing on DiffPure costs less than 1 day but it costs 2 days to test on our proposed method under the same setting with 8 V100 GPUs. Moreover, the number of paths in our method (Sec. 3.3) can be flexibly increased to be larger than two to greatly increase the time cost for attackers to generate adaptive attack examples. An accompanying merit is that the robust accuracy of our method in resisting DiffAttack is rather high because DiffAttack focuses on attacking the only one path by computing the gradient on it without meeting our dual path strategy.

Finally, our method outperforms the prior works with a gap in Table 5 remarkably larger than that in in Table 4. One main reason is due to the data size and the given random seeds between adaptive and non-adaptive attacks are quite different.

## 5 CONCLUSIONS & LIMITATIONS

We have presented a new test-time adversarial defense method. The key is to excessively denoise the incoming input image along the opposite adversarial path (OAP) so as to move far away from the decision boundary. This OAP prior can be readily plugged into the existing defense mechanisms for robustness improvement. Our defense method also forces attackers to spend a great deal of time creating adaptive adversarial examples. Meanwhile, we exemplify, for the first time, the pitfall of conducting AutoAttack (Rand) for diffusion-based adversarial defense methods. However, we are aware there are several attacks targeting diffusion-based adversarial defenses, and the performance of our proposed method may be overestimated since the gradient computation is approximated.

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
