REPRODUCIBILITY

To ensure reproducibility, we will make the source code publicly available after acceptance.

APPENDIX

## 6 RELATED WORKS: SUPPLEMENT

In Hill et al. (2020), the author proposed incorporating an energy-based model (EBM) with Markov Chain Monte Carlo (MCMC) sampling for adversarial purification. This method constitutes a memoryless sampling trajectory that removes adversarial signals, while this sampling behavior preserves image classes over long-run trajectories.

In Adaptive Denoising Purification (ADP) (Yoon et al. (2021)), the authors used the Noise Conditional Score Network (NCSN) with Denoising Score Matching (DSM) as the purifier, but with a deterministic short-run update rule for purification. This fixes the need for performing long-run sampling in order to remove adversarial noise in Hill et al. (2020).

Guided Diffusion Model for Purification (GDMP) (Wang et al. (2022), Wu et al. (2022)) is proposed to embed purification into the reverse diffusion process of a DDPM (Ho et al. (2020)). GDMP submerges adversarial perturbations with gradually added Gaussian noises during the diffusion process and removes both noises through a guided denoising process. By doing so, GDMP can significantly reduce the perturbations raised by adversarial attacks and improve the robustness of classification.

## 7 ADVERSARIAL ATTACKS

Two types of adversarial attacks are briefly introduced here.

**White-box attack.** The attacker knows all information about $f_\phi$, including the model architecture, parameters $\phi$, training schedule, and so on. One of the most indicative white-box attacks is projected gradient descent (PGD) (Madry et al. (2017)), a gradient-based attack. It produces adversarial perturbation by projecting NN's gradients on the clipping bound in an iterative manner. If the gradient is correctly calculated, the loss would certainly be maximized, and the NN, especially for non-robust NN, will be likely to return misclassified outputs.

**Black-box attack.** The attacker does not know all the information mentioned above in $f_\phi$. Common black box attack methods are roughly divided into two types: Query-base attack and Transfer-based attack.

No matter white- or black-box attack is concerned, in order to make adversarial perturbation less detectable, we will set a range of attack intensity, that is, adversarial perturbation is only allowed to perturb within a given norm value. Usually $\ell_p$-norm, denoted as $\|\cdot\|_p$ ($p = 1, 2, \infty$), is used:

$$\|\delta\|_p := \begin{cases} \left(\sum_i \delta_i^p\right)^{1/p} & p = 1 \text{ or } 2, \\ \max_i |\delta_i| & p = \infty. \end{cases} \tag{11}$$

In terms of intensity, we are accustomed to using $\epsilon_p$ to represent it. For example, it often uses $\epsilon_\infty = 8/255$ to indicate that the intensity range of currently used $\delta$ is $\|\delta\|_\infty \leq 8/255$.

## 8 DIFFUSION MODELS: SUPPLEMENT

Diffusion models were inspired by the diffusion phenomena under nonequilibrium thermodynamics in the physical world to design a framework that generates data by learning the reverse process of the data being destroyed by Gaussian noise gradually.

In the literature, the diffusion model (Sohl-Dickstein et al. (2015)) is a type of generative model (*e.g.*, GAN and VAE). Conceptually, this generative process behaves like denoising. Given a data point $x_0 \sim q$, where $q$ denotes the (unknown) true data distribution, and a variance schedule

$\{\beta_t\}_{t=1}^T$, the forward diffusion process follows $q(x_{1:T}|x_0) = \prod_{t=1}^T q(x_t|x_{t-1})$, where $q(x_t|x_{t-1}) = \mathcal{N}(x_t; \sqrt{1-\beta_t}x_{t-1}, \beta_t I)$, and the reverse diffusion process follows:

$$p_\theta(x_{0:T}) = p(x_T) \prod_{t=1}^T p_\theta(x_{t-1}|x_t), \tag{12}$$

where $p_\theta(x_{t-1}|x_t) \sim \mathcal{N}(x_{t-1}; \mu_\theta(x_t, t), \Sigma_\theta(x_t, t))$, $x_T \sim \mathcal{N}(0, I)$, and $\mu_\theta(x_t, t)$ and $\Sigma_\theta(x_t, t)$ denote the mean and covariance from the diffusion model parameterized by $\theta$ at time step $t$, respectively.

After that, there are several types of recently developed diffusion models, including score-based diffusion (Song & Ermon (2019); Song et al. (2020)), guided-diffusion (Dhariwal & Nichol (2021)), ILVR (Choi et al. (2021)), denoising diffusion probabilistic model (DDPM) (Ho et al. (2020)), and DDA (Gao et al. (2023)).

Specifically, in guided diffusion (Dhariwal & Nichol (2021)), given a label $y$ as the condition and Eq. (12), the conditional reverse process is specified as:

$$p_\theta(x_0, \ldots, x_{T-1}|x_T, y) = \prod_{t=1}^T p_\theta(x_{t-1}|x_t, y). \tag{13}$$

To solve Eq. (13), $\theta$ is decomposed into two terms as $\theta = \varphi \cup \phi$ to form separate models:

$$p_\theta(x_{t-1}|x_t, y) = Z p_\varphi(x_{t-1}|x_t) p_\phi(y|x_{t-1}), \tag{14}$$

where $Z$ is a normalization constant. Guided-diffusion improves the model architecture by adding attention head and adaptive group normalization, that is, adding time step and class embedding to each residual block. At the same time, with reference to GAN-based conditional image synthesis, class information is added during sampling and another classifier is used to improve the diffusion generator. To be precise, the pre-trained diffusion model can be adjusted using the gradient of classifier to direct the diffusion sampling process to any label.

Score-based diffusion (Song & Ermon (2019); Song et al. (2020)) generates samples by estimating the gradients of unknown data distribution with score matching, followed by Langevin dynamics, moving data points to areas with higher density of data distribution. In practice, the score network $s_\theta$ is trained to predict the true data distribution $q$ as:

$$s_\theta(x_t, t) \approx \nabla_{x_t} \log q(x_t) = -\frac{\epsilon_\theta(x_t, t)}{\sqrt{1-\bar{\alpha}_t}}, \tag{15}$$

where $\bar{\alpha}_t = \prod_{s=1}^t (1-\beta_s)$.

On the other hand, the conditional generation of the diffusion model has also received considerable attention. In ILVR (Choi et al. (2021)), the author proposed a learning-free conditioning generation, which is challenging in denoising diffusion probabilistic model (DDPM) (Ho et al. (2020)) due to the stochasticity of the generative process. It leveraged a linear low-pass filtering operation $\phi_N$ as a condition to guide the generative process in DDPM for generating high-quality images based on a given reference image $c$ at time $t$, termed $c_t$, which can be obtained by the forward diffusion process $q$. The update rules are derived as follows:

$$\begin{aligned} x'_t &\sim p_\theta(x'_t|x_{t+1}) \\ c_t &\sim q(c_t|c) \\ x_t &\leftarrow \phi_N(c_t) - x'_t - \phi_N(\hat{x}'_t), \end{aligned} \tag{16}$$

where the factor of downsampling and upsampling is denoted as $N$.

DDA (Gao et al. (2023)) also came up with a similar approach to resolve the domain adaptation problem in the test-time scenario. The authors adapt the linear low-pass filtering operation $\phi_N$ in ILVR (Choi et al. (2021)) as conditions, and their method also forces the sample $x_t$ to move in the direction that decreases the distance between the low-pass filtered reference image $\phi_N(x_0)$ and low-pass filtered estimated reference image $\phi_N(\hat{x}_0)$. The update rule is specified as follows:

$$\hat{x}_0 \leftarrow \sqrt{\frac{1}{\bar{\alpha}_t}} \, x_t - \sqrt{\frac{1}{\bar{\alpha}_t} - 1} \, \epsilon_\theta(x_t, t) \tag{17}$$

$$x_t \leftarrow \hat{x}_t - w \nabla_{x_t} \|\phi_N(x_0) - \phi_N(\hat{x}_0)\|_2, \tag{18}$$

where $N$ is the factor of downsampling and upsampling, and $w$ is the step size.

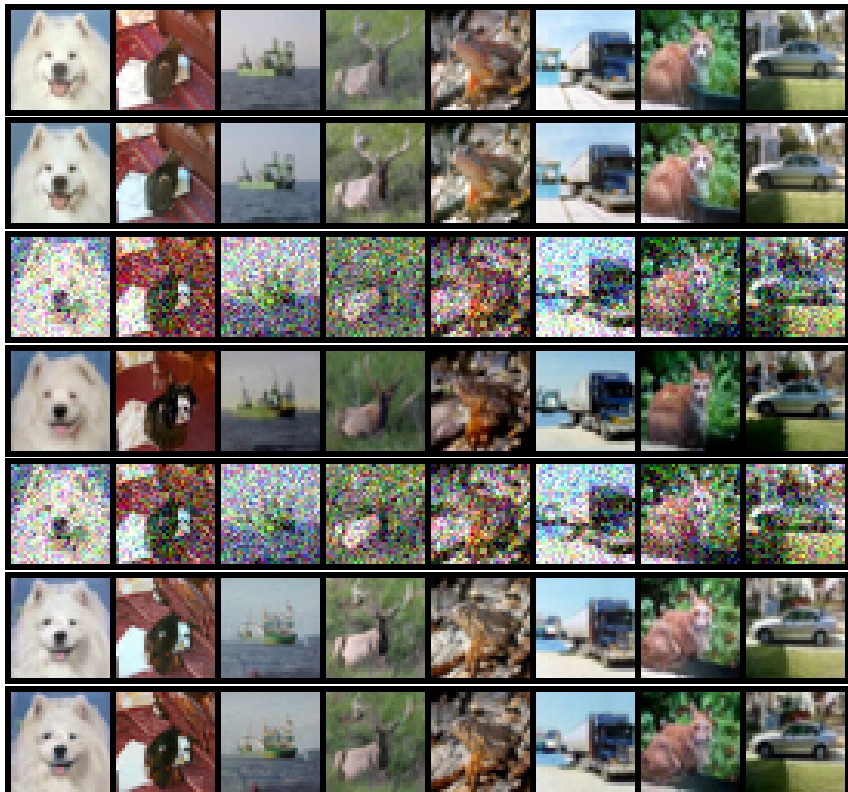

Figure 5: Intermediate Images generated from Fig. 2(c). From Top to Bottom: The images denote clean image $x$, $x^{p_2}$, $x_{t^*}^{p_1}$, $\widehat{x^{p_1}}$, $x_{t^*}^{p_2}$, $\widehat{x^{p_2}}$, and purified image $\widehat{x_{clean}}$, respectively.

## 9 INTERMEDIATE IMAGES GENERATED FROM FIG. 2(C)

In Fig. 5, we show the images generated from each step in Fig. 2(c) for visual inspection.

## 10 ATTACK COST IN TIME COMPLEXITY: DETAILED ANALYSIS

In this section, we discuss the cost the attackers need to pay to defeat our proposed test-time adversarial defense method. In particular, we focus on analyzing the time complexity of defeating the diffusion-based purifiers presented in Sec. 3.2 and Sec. 3.3.

In DISCO (Ho & Vasconcelos (2022)), let $N_d$ and $N_c$ be defined as the number of parameters in DISCO and its downstream classifier, respectively. In *training time*, the authors estimated that the time complexity of defense/purification is $\mathcal{O}(N_d)$ and that of adaptive attack (*i.e.*, designing an adversarial example by knowing the whole model) is $\mathcal{O}(KN_d + N_c)$, where $K$ is the number of steps. Hence, the ratio of attack-to-defense cost in training time is $\mathcal{O}(K + N_c/N_d)$ and the authors asserted that the time complexity of defense is much less than that of attack since $N_d < N_c$. In *testing time*, the time complexity of defense becomes $\mathcal{O}(KN_d)$ (see green bars in Fig 10 of Ho & Vasconcelos (2022)) while the other remains the same. Hence, the ratio of attack-to-defense cost in testing time is $\mathcal{O}(1 + N_c/(KN_d))$, revealing that even $N_c$ is larger than $N_d$, the time complexity for both the attacker and defender can tie if $K$ is large enough.

In DiffPure (Nie et al. (2022)), the authors argued that an adaptive attack on diffusion model by traditional back-propagation would cause high memory cost. To overcome this issue, they instead applied *adjoint method* (Li et al. (2020)) to efficiently estimate the gradient used for designing adaptive adversarial perturbation under constant memory cost. From Table 1 in Li et al. (2020), it is asserted that, under the *tolerance* $\epsilon = 1/T$, the per-step time complexity scales as $\log_2 T$, where $T$ is

the number of steps required during the reversed process. So, the time complexity of adjoint method through the reverse diffusion process is $\mathcal{O}(T \log_2 T)$.

We, however, raise the concern that the presented time cost is likely to ignore one important factor: the number of parameters in a diffusion model, denoted as $N_{dm}$. Hence, in conservatively speaking, the time complexity of adjoint method through the reverse process should be corrected as $\mathcal{O}(N_{dm} T \log_2 T)$.

Based on the above concerns, our estimations of the time complexity of adversarial attack and proposed defense methods in *testing time* are described as follows. For the defense method described in Sec. 3.2, since the baseline purifier $g_\theta$ is reused in every step of the diffusion process, $T$ is equal to $K$ (hereafter, we will use them interchangeably). So, the time complexity costs of adaptive attack and our defense are derived as $\mathcal{O}(N_{dm} T \log_2 T + T N_d + N_c)$ and $O(N_{dm} T + N_d T + N_c)$, respectively. In practice, $N_{dm}$ is usually much larger than $N_d$ and $N_c$, so we have $\mathcal{O}(N_{dm} T \log_2 T + T N_d + N_c) \approx \mathcal{O}(N_{dm} T \log_2 T)$ and $\mathcal{O}(N_{dm} T + N_d T + N_c) \approx \mathcal{O}(N_{dm} T)$, and the ratio of attack-to-defense cost is $\mathcal{O}(\log_2 T)$. Moreover, if the attacker adopts the Expectation Over Time (EOT) operation with a number of iterations, $T_{EOT}$, the time cost of creating such an attack has to be additionally multiplied by $T_{EOT}$. For example, if $T = 100$ and $T_{EOT} = 20$, the ratio theoretically approximates 133; *i.e.*, the time cost of adaptive attack is around 133 times as big as that of defense.

For the defense method described in Sec. 3.3, since two diffusion paths should be maintained during purification, the time complexity is simply doubled than the one in a single path (Sec. 3.2). It is concluded that although the inference time of our proposed test-time defense methods is increased, the increased cost also complicates the adaptive attacks as well.

Table 6 shows the time cost comparison between the reverse diffusion process (purification) and adaptive attack (BPDA+20 EOT), implemented using the adjoint strategy in Sec. 3.4, under DiffPure and our defense methods. Here, we do not consider the adaptive AutoAttack since the mechanism is much more complex beyond the above analysis.

| Methods | Purification Time (sec.) | Attack Time (sec.) | Ratio |
|---|---|---|---|
| DiffPure (Nie et al. (2022)) | 35.20±0.67 | 2508.56±116.06 | 71.27 |
| Sec. 3.2 | 34.59±0.95 | 2504.14±112.61 | 72.39 |
| Sec. 3.3 | 72.84±34.41 | 7517.03±1970.35 | 103.20 |

Table 6: Computational time cost comparison in five runs. This evaluation utilizes four testing images with batch size two, running on one NVIDIA V100. The ratio is calculated as the average time required for BPDA+20 EOT divided by that for reverse diffusion.

## 11 MORE EXPERIMENTAL RESULTS

In this section, we examine the robustness of our method using CIFAR-100 Krizhevsky et al. (2009) and ImageNet Deng et al. (2009). Due to limited computation resources and budgets, some attacks on these datasets are still running. We hope that in the near future, the results can be ready since they are extremely eager for computation resources.

### 11.1 RESULTS ON CIFAR-100

We provide the robustness evaluation against adversarial attacks on CIFAR-100, as shown in Table 7 (*cf.* Tables 4 and 5 for CIFAR-10). Similarly, the experiments in the first block of Table 7 are under the setting of non-adaptive attack, in which the attacker only knows the information of the downstream classifier. We also excerpt the results of Rebuffi et al. (2021); Wang et al. (2023); Cui et al. (2023) from RobustBench (Croce et al. (2021)) for more comparisons. The second block of Table 7 shows the results obtained under the setting of adaptive attacks. Note that the results for DiffPure (Nie et al. (2022)) are from Zhang et al. (2024). We can find that our methods are either better than the prior works under BPDA+EOT or comparable with DiffPure under AutoAttack and PGD-$\ell_\infty$.

| Defense Methods | Clean Acc (%) | Robust Acc (%) | Attacks |
|---|---|---|---|
| No defense | 81.66 | 0 | PGD-$\ell_\infty$ |
| Rebuffi *et al.* (Rebuffi et al. (2021)) | 62.41 | 32.06 | AutoAttack (Standard) |
| Wang *et al.* (Wang et al. (2023)) | 72.58 | 38.83 | AutoAttack (Standard) |
| Cui *et al.* (Cui et al. (2023)) | **73.85** | 39.18 | AutoAttack (Standard) |
| DiffPure (Nie et al. (2022)) | 61.96±2.26 | 59.27±2.95 | PGD-$\ell_\infty$ |
| DiffPure (Nie et al. (2022)) | 61.98±2.47 | **61.19±2.87** | AutoAttack (Standard) |
| Ours (Sec. 3.2) | 61.71±2.49 | 60.21±1.82 | PGD-$\ell_\infty$ |
| Ours (Sec. 3.2) | 62.02±2.24 | 60.08±2.44 | AutoAttack (Standard) |
| No defense | 81.66 | 0 | BPDA+EOT |
| DiffPure (Nie et al. (2022); Zhang et al. (2024)) | 69.92 | 48.83 | BPDA+EOT |
| Hill *et al.* (Hill et al. (2020))* | 51.66 | 26.10 | BPDA+EOT |
| ADP ($\sigma = 0.1$) (Yoon et al. (2021))* | 60.66 | 39.72 | BPDA+EOT |
| Ours (Sec. 3.3) | **70.38±4.05** | **51.25±3.82** | BPDA+EOT |

Table 7: Robustness evaluation and comparison between our method and state-of-the-art methods. Classifier: WRN-28-10. Testing dataset: CIFAR-100. Asterisk (*) indicates that the results were excerpted from the papers. Boldface indicates the best performance for each attack.

## 11.2 RESULTS ON IMAGENET

We provide the robustness evaluation against non-adaptive PGD-$\ell_\infty$ on ImageNet, as shown in Table 8. This experiment was conducted on a more advanced transformer-based classifier (Liu et al. (2022)) with $\|\delta\|_\infty \leq 8/255$ and ResNet-50 with $\|\delta\|_\infty \leq 4/255$. Our method proposed in Sec. 3.3 is slightly better than DiffPure, while Our method proposed in Sec. 3.2 is comparable with DISCO, which, however, needs additional data for training EDSR for purification.

| Defense Methods | Clean Acc (%) | Robust Acc (%) | Classifier | Attack |
|---|---|---|---|---|
| DISCO (Ho & Vasconcelos (2022))* | **72.64** | 66.32 | ResNet-50 | PGD-$\ell_\infty$ (4/255) |
| Ours (Sec. 3.2) | 69.32±12.18 | **68.12±12.13** | ResNet-50 | PGD-$\ell_\infty$ (4/255) |
| Ours (Sec. 3.3) | 67.55±11.73 | 66.54±12.15 | ResNet-50 | PGD-$\ell_\infty$ (4/255) |
| DiffPure (Nie et al. (2022)) | 75.13±11.67 | 73.11±11.76 | SwinV2 (Liu et al. (2022)) | PGD-$\ell_\infty$ (8/255) |
| Ours (Sec. 3.2) | 75.37±12.01 | 71.73±12.63 | SwinV2 (Liu et al. (2022)) | PGD-$\ell_\infty$ (8/255) |
| Ours (Sec. 3.3) | **75.38±12.78** | **73.20±11.27** | SwinV2 (Liu et al. (2022)) | PGD-$\ell_\infty$ (8/255) |

Table 8: Robustness comparison between our method and DISCO/DiffPure. Testing dataset: ImageNet. Asterisk (*) indicates that the results were excerpted from the paper. Boldface indicates the best performance for each attack.

## 12 ALGORITHM IN SEC. 3.3

Here, we describe the entire procedure of the proposed method in Sec. 3.3.

---

**Algorithm 1** Diffusion Path Cleaning-based Purifier

---

**Require:** Purifier $g_\theta$, adversarial image $x_{adv}$
**Ensure:** Purified image $\widehat{x_{clean}}$
1: $x \leftarrow x_{adv}$
2: **for** repeated time from $1 \ldots 2$ **do**
3:     $x^{p_1} \leftarrow x; x^{p_2} \leftarrow x$
4:     $j \leftarrow \underset{i \in \{1,\ldots,C\}}{\operatorname{argmin}} \ S_\varepsilon(x^{p_2}, x_i^{tar})$
5:     $x^{p_2} \leftarrow f_{CT}(x^{p_2}, x_j^{tar})$
6:     $x^{p_1} \leftarrow g_\theta(x^{p_1}); x^{p_2} \leftarrow g_\theta(x^{p_2})$
7:     $x \leftarrow f_{CT}(x^{p_2}, x^{p_1})$
8: **end for**
9: $\widehat{x_{clean}} \leftarrow x$
10: **return** $\widehat{x_{clean}}$

---