# OpenReview forum: "Test-Time Adversarial Defense with Opposite Adversarial Path and high Attack time cost"
_ICLR.cc/2025/Conference — Submitted to ICLR 2025_

### Official Review · Reviewer_xpXC · 2024-10-29

**Soundness:** 2
**Presentation:** 2
**Contribution:** 3
**Rating:** 5
**Confidence:** 4

**Summary:**

This paper introduces a test-time adversarial defense method, which exploits the diffusion-based recovery through opposite adversarial paths (OAPs) to enhance robustness against adversarial attacks. This work proposes denoising along the opposite adversarial gradient and incorporating it with reverse diffusion. Experiments on three datasets are conducted to validate the effectiveness of the proposed method.

**Strengths:**

1. This paper introduces a new method to alleviate the influence of adversarial noise, which proposes to leverage the opposite adversarial path and reverse diffusion.

2. The experiments are performed on three popular datasets, which provides extensive evaluation results.

3. Multiple related works are considered for comparison to present the gains brought by the proposed method.

**Weaknesses:**

1. The statement in Section 3.1.2 is contradictory to the statement in Section 1.3, and the corresponding explanation is not convincing enough.

2. In this work, the proposed method only consider the manner of non-targeted adversarial attacks. For the targeted attacks, the adversarial gradient is varied with the specified target class, and the proposed method with a single opposite adversarial path seems not to be able to handle this case.

3. The adversarial attacks used to evaluate the defenses mainly belong to gradient-based non-targeted methods, the optimization-based adversarial attacks (e.g., C&W, DDN), targeted attacks and black-box attacks can be considered to more comprehensively evaluate the effectiveness of the proposed method.

4. Some details of the experiment are unclear.

5. Some typos, for example, "To this end, according to (14) of guided diffusion described in Sec. 8 in Supplementary" (Line 264) → "To this end, according to Eq.(14) of guided diffusion described in Sec. 8 in Supplementary"

**Questions:**

1. The statement in Section 3.1.2 is contradictory to the statement in Section 1.3, and the corresponding explanation is not convincing enough. In Section 1.3, the clean accuracy decreases at K=1, and the authors argue that “the process ‘-adv’ is still unstable, hence, some data points near the decision boundary may be perturbed to incorrect class”, but in Section 3.1.2, the model has better performance at K=1.  Although the authors conjecture that this inconsistency is because the ground-truth labels are not used in Table 2, the training data $x^K$ is actually generated step-by-step using the ground-truth label y as well, and why this is not unstable. The authors can clarify the specific differences in methodology or experimental setup between these two terms or provide a more detailed analysis of the stability of the '-adv' process across different values of K

2. Some details of the experiment are unclear. In Tables 1-5, it is not clear whether the adversarial attack is against the target model only or against the target model and purifier as a whole. It seems that the described adaptive attack is against the target model only by using BPDA, EOT, etc., rather than a full white-box attack (e.g., C&W) against the target model and the purifier as a whole. The authors can explicitly state for each experiment whether the adversarial attacks are targeting only the model or both the model and purifier.

---

> ### Author Response · Authors · 2024-11-20
>
> **Response to W1/Q1 (Regarding statements in Sec. 3.1.2 and Sec. 1.3):** The difference between the two experimental setups is as follows.
> * For our Table 1, in the process $x_{adv} \to x \to x^1 \to \cdots \to x^K$, $x^i$ will gradually converge to the area, where the classifier best fits (because the ground-truth label is known). Even though the process may be unstable at some small $K$ (e.g. $K=1$), corresponding to the drop of clean accuracy of the classifier, it will eventually fit the classifier perfectly as $K$ increases.
> * For our Table 2, the performance of the purifier that maps $x_{adv}\to x^K$ depends on the step size for the optimization step. When $K$ is small (e.g. $K=1$), the difference between $x_{adv}$ and $x^K$ is small, so the step size is small, making the purifier trained better, and hence the downstream classifier will have better performance.
>
> For more information, we respond from three perspectives.
>
> * First, we draw reviewers’ attention to Table 1, where the experiments were conducted on the test dataset under the following conditions:
>     1. We first attack the input with the adversarial attack and then move the attacked input with $-K$ steps while the label $y$ is known at every step.
>     2. The direction computed at each step is a gradient direction toward a lower loss area.
>     3. The data are moved as the form of $x_{adv} \to x \to x^1 \to \cdots \to x^K$ in a "step by step" manner.
>
>     This experiment aims to explore how the opposite gradient direction influences robust accuracy and, more importantly, to see if a safe zone exists for each data point. To utilize such gradient information surrounding the decision boundary, we proposed to learn the mapping $x_{adv} \to x^K$ by an NN model $g_\theta(\cdot)$. In other words, we want to use $g_\theta(\cdot)$ to approximate $K$ gradient steps.
> * On the other hand, the experiment in Table 2 was also conducted on the test dataset but with a learned $g_\theta(\cdot)$ under the conditions different from those in Table 1:
>     1. We first attack the input with the adversarial attack.
>     2. $g_\theta(\cdot)$ learns from $x_{adv} \to x^K$ only in one step, i.e., $x^K=g_\theta(x_{adv})$.
>     3. In testing stage, label $y$ is NOT given while we move the input with $g_\theta(\cdot)$.
>
>     Since label $y$ is unknown during inference, the goal of training $g_\theta$ is to fit the path from adversarial examples to an excessively denoised data via $K$-step. From the perspective of gradient descent, the step size is crucial to finding a better point for the next optimization step. When the step size is small, the start and end points approximate the linearity of the objective function. Connecting these small steps will form a curve or a zigzag path.  Such paths are non-linear but they describe complicated gradient information.
>
>   Yet, if one chooses a large step size, the algorithm might overshoot the better endpoint since a large step size ignores such detailed information around the endpoint (e.g. a zigzag path). In our explanation for Table 2 in the paper, we analogize such $x_{adv} \to x^K$ as a large step size in gradient descent when $K>1$. Therefore, it is too coarse for $g_\theta(\cdot)$ to capture the details of gradient information around the decision boundary if we use $x^K$, where $K>1$, as the endpoint for training a $g_\theta(\cdot)$. Hence, the large step effect and the lack of label guidance lead to decreased performance in Table 2 while $K$ increases.
> * In Table 1, due to the presence of $y$ and step-by-step update, we can find useful gradient information in every step. Therefore, using a purifier that maps $x_{adv} \to x^1$ (i.e. with a small step) and repeatedly applying such an update during the inference is a better way to capture gradient information. That is why we adopt this strategy alongside the reverse diffusion process of score-based diffusion for moving the adversarial data to a safer area.
>
> **Cont.**

---

> ### Author Response · Authors · 2024-11-20
>
> **Response to W2/W3 (Regarding the optimization-based adversarial attacks (e.g., C&W, DDN), targeted attacks and black-box attacks):** We will respond to W2/W3 together from three aspects below!
> * Actually, we have to clarify that for a fair comparison with SOTA works, we have already verified our method with attacks tested by known methods.
>     Specifically, the well-known BPDA+EOTs, PGD+EOTs [R1-1], and DiffAttack [R2-6] are already optimization-based adversarial attacks in that the gradient computation, under our implementation described in Sec. 3.4, will go through every module from the input to output, including all iteratively used UNet, DISCO, color OT in purifier, and downstream classifiers (results are in our Table 5).
> * As for DDN [R3-10], it actually only considers $\ell_2$ attack, which is weaker than $\ell_\infty$ attack in general. Please refer to the performance gap among those on Robustbench [R1-4] for details. Moreover, as mentioned in [R1-3], "The black-box threat model is a strict subset of the white-box threat model, so attacks in the white-box setting should perform better; if a defense is obfuscating gradients, then black-box attacks (which do not use the gradient) often perform better than white-box attacks," which the authors of [R1-3] refer to [R3-8].
> * Finally, in [R2-6], the authors also compared DiffAttack with other attacks (Table 4 in [R2-6]), showing that DiffAttack outperforms other attacks. As for targeted attacks, we have already tested our methods on AutoAttack (in our Table 4), including untargeted APGD-CE, targeted APGD-DLR, targeted FAB, and Square Attack as listed on their GitHub page [R3-9]. According to [R2-7], most of the time, untargeted attacks outperform targeted attacks in ASR. Furthermore, we have thoroughly surveyed the literature, and to our knowledge, we have not found any targeted adversarial attack on diffusion-based purification. If the reviewer knew these works, could you please provide us? We are happy to read and respond to you!
>
> **Response to W4/Q2 (unclear details of experiments):** We respond to this comment from three aspects.
> * First, we draw the reviewer's attention to Lines 461 $\sim$ 462, where we claimed that "To the most extreme...the attacker knows every detail about our framework of "purifier+classifier,"... we utilize BPDA to ... ". The computation of BPDA is actually done with the adjoint method, which approximates the gradients of the diffusion reverse process. This is also the attack tested in DiffPure and other related works. However, this attack has been criticized by [R1-1] in that "When calculating gradients, it is best to directly back-propagate the full defense process. If this is unavailable due to memory constraints, using the surrogate process rather than the adjoint method is recommended." Hence, to avoid overestimation, we followed the same strategy and tested PGD+EOTs [R1-1] and DiffAttack [R2-6] in a white box setting.
> * Second, combined with our findings in Sec. 3.4, our implementation allows the above attacks, including BPDA, to go through every module from the input to output, including all iteratively used UNet, DISCO, color OT in purifier, and downstream classifiers. The results are presented in Table 5. It should be noted that Table 1 represents ONLY the non-defense classifier against PGD with $-K$ steps and Table 2 presents two non-adaptive attacks, which only know the classifier, and one BPDA adaptive attack on purifier+classifier. These should also have been clearly revealed in the tables and captions. In addition, Table 3 demonstrates that our discovery and implementation can turn AutoAttack (Rand) into a stronger attack that adapts to diffusion-based purification.
> * Third, for the attacks in Table 4, as revealed in its captions, are non-adaptive and only see the classifiers' information. Thanks for your understanding! If there are still something unclear, the authors are happy to clarify per your questions!
>
> **Response to W5 (typos):** We have revised the typos (see revised PDF).
>
> **Refs:** \
> [R1-1] Robust Evaluation of Diffusion-Based Adversarial Purification, ICCV 2023. (Lee & Kim (2023) in our paper) \
> [R1-3] Obfuscated Gradients Give a False Sense of Security: Circumventing Defenses to Adversarial Examples, ICML 2018. (Athalye et al. (2018) in our paper) \
> [R1-4] https://robustbench.github.io/ (Croce et al. (2021) in our paper) \
> [R2-6] DiffAttack: Evasion Attacks Against Diffusion-Based Adversarial Purification, NeurIPS 2023. (Kang et al. (2024) in our paper) \
> [R2-7] Diversity can be transferred: Output diversification for white-and black-box attacks, NeurIPS 2020. \
> [R3-8] Practical black-box attacks against machine learning, ASIA CCS 2017. \
> [R3-9] https://github.com/fra31/auto-attack (Croce & Hein (2020) in our paper) \
> [R3-10] Decoupling direction and norm for efficient gradient-based l2 adversarial attacks and defenses, CVPR 2019.

---

> > ### Comment · Reviewer_xpXC · 2024-11-26
> > **Response to authors' rebuttal**
> >
> > Thank authors for their responses and explanations. A number of explanations for the two experimental setups are provided, but I think the organization and description in the main text would still be confusing to the reader. Besides, the evaluations against representative optimization-based attacks (e.g. C\&W) are not presented, the defense that are effective against AA are not necessarily effective against L2-norm optimization-based attacks. As for the targeted attacks against the diffusion-based purification, it can be autonomously designed or even crafted as adaptive attacks to evaluate defenses, for example, setting the optimization goal to generate a specified sample or maximizing the mean-square error between the generated and original samples. Thus, I will keep the original rating score.

---

> ### Author Response · Authors · 2024-11-29
>
> We sincerely thank you for your feedback!
> 1. In responding to the Reviewer's comment, we have conducted a new experiment of defense against C&W (L2-norm attack). Please see Tables I and II below. The experimental results in Table I show that under a non-adaptive setting, our method performs the same in resisting against AA (Standard) and C&W (L2-norm attack), revealing its effectiveness in dealing with both AA and L2-norm optimization-based attacks (e.g., C&W).
> 2. "As for the targeted attacks against the diffusion-based purification, ... the generated and original samples." In responding to this comment, we implemented such an attack to verify our method in Table II. It can be seen that our method still performs very well and PGD+EOT, which was tested before, seems to be stronger than other attacks. However, we have to clarify that such a targeted attack, **as suggested by the reviewer**, is a special case of DiffAttack since it only utilizes the input image and purified image to optimize the attack direction, in contrast to the DiffAttack that utilizes outputs of **every single step** in the diffusion forward and reverse processes, which is indicated in Eq. (8) of [R2-6] as:
> $$\max \mathcal{L_{dev}}=E_t[\alpha(t)E_{x_t,x_t'|x_0}d(x_t, x_t')]$$
> where $\alpha(\cdot)$ is time-dependent weight coefficients, and $d(x_t, x_t')$ is the distance between original image $x_t$ in the diffusion process and generated image $x_t'$ in the reverse process at time $t$, and presented in their implementation [R2-9]. \
> The objective of the attack is as follows:
> $$\max\mathcal{L}=\mathcal{L_{cls}}+\lambda\mathcal{L_{dev}}$$
> where $\lambda$ is the weight coefficient. For the $\mathcal{L}_{cls}$, we also set it to a **targeted attack loss**.
> 3. For a thorough evaluation, we also present the results on SPSA [R2-11], **which uses gradient-free optimization techniques**, under both settings (non-adaptive/adaptive).
>
> In addition, the gradient computation from PGD+EOT and DiffAttack coincides with the recommendations from [R1-1] and [R2-10], stating "*When calculating gradients, it is best to directly back-propagate the full defense process,* and *always try to implement adaptive attacks that are specific to adaptive defenses*, respectively."
>
> Table I. Non-adaptive attacks on Sec. 3.2. Note that results of PGD-$\ell_\infty$ and AA are from Table 4 in our paper (attack on classifier).
> |Attacks|Clean Accuracy (%)|Robust Accuracy (%)|
> |:-|:-:|:-:|
> |PGD-$\ell_\infty$|90.77±2.25|88.48±2.04|
> |AA (Standard)| 90.46±2.36 | 89.06±2.62 |
> |C&W|90.62±1.70|90.56±1.51|
> |SPSA [R2-11]|90.46±1.91|90.65±1.83|
>
> Table II. Adaptive attacks on Sec. 3.3. Note that results of PGD+EOT and DiffAttack are from Table 5 in our paper (attack on purifier+classifier).
> |Attacks|Clean Accuracy (%)|Robust Accuracy (%)|
> |:-|:-:|:-:|
> | PGD+EOT | 100 | 53.12 |
> | DiffAttack | 95.31 | 93.75 |
> | Targeted attack |95.31|95.31 |
> |C&W| 90.17±4.31 | 85.09±6.37 | 0
> |SPSA [R2-11]| 89.32±4.85 | 89.19±4.81 |
>
> [R1-1] Robust Evaluation of Diffusion-Based Adversarial Purification, ICCV 2023. (Lee & Kim (2023) in our paper) \
> [R2-6] DiffAttack: Evasion Attacks Against Diffusion-Based Adversarial Purification, NeurIPS 2023. (Kang et al. (2024) in our paper) \
> [R2-9] https://github.com/kangmintong/DiffAttack/blob/main/DiffAttack_Score_Based/diffattack/diffattack_base.py#L213 \
> [R2-10] Evaluating the adversarial robustness of adaptive test-time defenses, ICML 2022. (Croce et al. (2022) in our paper) \
> [R2-11] Adversarial risk and the dangers of evaluating against weak attacks, ICML 2018.

---

### Official Review · Reviewer_vNum · 2024-10-31

**Soundness:** 2
**Presentation:** 3
**Contribution:** 3
**Rating:** 5
**Confidence:** 4

**Summary:**

This paper proposes a preprocessing-based test time defense against adversarial attacks. The proposed method relies on the minimization of training loss and adversarial purification using the diffusion model. The adversarial inputs are pushed towards the opposite adversarial direction via the reverse diffusion process. The proposed purifier can be plugged into a pre-train model, which means the proposed method is easily combined with different types of defenses, such as adversarial training.

**Strengths:**

- The idea of reducing the effects of adversarial perturbations by perturbing the input in a way that minimizes training loss makes sense. This process pushes the adversarial input to non-adversarial area. This idea cannot be applied directly because because the ground truth label is not available during inference. The proposed method is able to emulate this process through an inverse diffusion process and achieve efficient denoising.
- The effectiveness of the proposed method is empirically demonstrated by the experimental results. It also shows superior results compared to existing methods against the strong attacks such as EOT and BPDA , which are designed to defeat randomized defenses.

**Weaknesses:**

- To demonstrate that the proposed method is effective against various attacks, it would be better to experiment with methods that converge to adversarial examples that differ from PGD and APGD, such as ACG and PGD-ODI. This is because PGD-based attacks highly depend on the initial point of the attack.
The effectiveness of random initialization, such as Output Diversified Sampling (ODS) in adversarial attacks, implies the high dependency of PGD-based attacks on initial points. It is natural to assume the same dependency in the minimization of training loss with respect to input because PGD-based attacks maximize the same loss to generate adversarial examples.
However, the attacks used for training and evaluation of the proposed method are all derivatives of PGD. Thus, the adversarial examples subject to purification are likely to be similar to each other.
- It would be helpful if you could explain the difference between the solid and dashed lines in the arrows in the figures.

**Questions:**

- Does the proposed method show superior performance against attacks, such as PGD-ODI and ACG,  that converge to different adversarial examples from PGD?
- What do the differences in types and colors of lines mean in the figures?

---

> ### Author Response · Authors · 2024-11-20
>
> **Response to W1/Q1 (Regarding ACG and PGD-ODI):** We respond to this comment in four aspects.
> * First, for a fair comparison with other works, we verified our method with attacks tested by known methods.
> * Second, the authors of ACG [R2-5] mentioned in Appendix M that they did not compute the attack success rate (ASR) between ACG and AutoAttack because the authors claimed that "...because we focused on generating many adversarial examples quickly." Also, based on the fact that the ASR difference between ACG and APGD [R2-5] and that between PGD-ODI and PGD [R2-7] are both smaller than that between AutoAttack and PGD+EOTs in [R1-1] (in Table 7, AP section), so we prioritized testing PGD+EOTs in our evaluations (in our Table 5) in a more accurate manner concerning gradient computation (see Sec. 3.4 of our paper).
> * Third, we realized that the uprising diffusion-based purification is harder to attack, so we not only tested our method on PGD+EOTs but also the DiffAttack [R2-6] especially targeted diffusion-based purification in attacking intermediate output at each reverse step (in our Table 5). More importantly, in [R2-6], the authors also compared DiffAttack and other attacks (in Table 4), and DiffAttack was demonstrated to exhibit better performance than other attacks.
> * Finally, the evaluation in our Table 5 was conducted under the white-box settings, which means all the iteratively used UNet, DISCO, our color OT in the purifier, and the downstream classifier are known to the attacker.
>
> **Response to W2/Q2 (Difference between the solid and dashed lines in the arrows in the figures):** We guess the reviewer refers to Figure 2? If so, the dashed arrow means that $\widehat{x_{clean}}$ is reused as the input to dual-path purification. The solid lines mean the purification direction in which several modules, such as forward & reverse diffusion, and Color OT, will process the data. Besides, as indicated in the caption of Figure 2, we have described that  "The image in front of Color OT with green/blue arrow is called the source/target image."
>
> **Refs:** \
> [R1-1] Robust Evaluation of Diffusion-Based Adversarial Purification, ICCV 2023. (Lee & Kim (2023) in our paper) \
> [R2-5] Diversified adversarial attacks based on conjugate gradient method, ICML 2022. \
> [R2-6] DiffAttack: Evasion Attacks Against Diffusion-Based Adversarial Purification, NeurIPS 2023. (Kang et al. (2024) in our paper) \
> [R2-7] Diversity can be transferred: Output diversification for white-and black-box attacks, NeurIPS 2020.

---

> > ### Comment · Reviewer_vNum · 2024-11-27
> >
> > Dear Authors,
> > Thank you for your detailed responses to my comments.
> > I have carefully reviewed your replies. The selection of attack baselines used in your experiments appears reasonable, considering the computational costs for the evaluation. However, my primary concern remains that the proposed method could be potentially vulnerable to certain existing attacks. To strengthen the claims, I think the proposed method should be tested against a broader range of attacks, especially those specifically targeting adversarial purification techniques.
> > Based on these considerations, I have decided to retain my original score.

---

> ### Author Response · Authors · 2024-11-29
>
> We sincerely thank you for your feedback!
> 1. First of all, the attacks used in evaluating our method are  **not baseline** attacks. Please note that PGD+EOT [R1-1] and DiffAttack [R2-6] are by far the strongest and most effective attacks, **especially targeting diffusion-based adversarial purification**. They are those that the reviewer mentioned **certain existing attacks**. Additionally, Tables 1 and 3 in [R1-1] show that PGD+EOT is a general and effective attack against purification-based methods. Thanks for your understanding!
> 2. Second, the reason we tested those attacks is not mainly due to consideration of computational cost but for exploring the robustness against the **strongest attacks**. Recall that we proposed in Sec. 3.4 (GRANULARITY OF GRADIENT APPROXIMATION IN REALIZING POWERFUL ADAPTIVE AUTOATTACK) of discussing how to implement and resist a stronger adaptive attack. This merit has also been recognized by Reviewer wQgS. Thanks for your understanding that we aim to test on SOTA attacks. Besides, the attackers' computational cost is increased because of our design, and we provided such information along with the SOTA robust accuracy to be one of the contributions of our method.
> 3. Third, based on the following prior results, including (1) the attack success rate (ASR) difference between DiffAttack and other attacks, which was shown in Table 4 [R2-6], and (2) the conclusion made by [R1-1] and [R2-10], where *"When calculating gradients, it is best to directly back-propagate the full defense process"* in the **Recommendation** section on Page 5 of [R1-1] and *"always try to implement adaptive attacks that are specific to adaptive defenses... [R2-10],"* we indeed have already presented the robust accuracy against the SOTA attacks (PGD+EOT and DiffAttack) in Table 5 of our paper.
>
> BTW, we have also conducted new experiments on some attacks according to Reviewer xpXC's suggestions, including (1) C&W attack, (2) adaptive targeted attack, and (3) SPSA [R2-11], which used **gradient-free optimization techniques**. As the Reviewer xpXC commented: "As for the targeted attacks against the diffusion-based purification, ... the generated and original samples." In responding to this comment, we implemented such an attack to verify our method in the following Table I. It can be seen that our method still performs very well and PGD+EOT, which was tested before in our submission, seems to be stronger than other attacks. However, we have to clarify that such a targeted attack, **as suggested by the reviewer xpXC**, is a special case of DiffAttack (which we have verified in our paper) since it only utilizes the input image and purified image to optimize the attack direction, in contrast to the DiffAttack that utilizes the output of **every single step** in the diffusion forward and reverse processes, which was indicated in Eq. (8) of [R2-6] as:
> $$\max \mathcal{L_{dev}}=E_t[\alpha(t)E_{x_t,x_t'|x_0}d(x_t, x_t')]$$
> where $\alpha(\cdot)$ is time-dependent weight coefficients, and $d(x_t, x_t')$ is the distance between original image $x_t$ in the diffusion process and generated image $x_t'$ in the reverse process at time $t$, and presented in their implementation [R2-9]. \
> Moreover, the objective of this attack is as follows (Eq. (11) of [R2-6]):
> $$\max\mathcal{L}=\mathcal{L_{cls}}+\lambda\mathcal{L_{dev}}$$
> where $\lambda$ is the weight coefficient. For the $\mathcal{L}_{cls}$, we also set it to a **targeted attack loss**.
>
> Table I. Adaptive attacks on Sec. 3.3. Note that results of PGD+EOT and DiffAttack are from Table 5 in our paper (attack on purifier+classifier).
> |Attacks|Clean Accuracy (%)|Robust Accuracy (%)|
> |:-|:-:|:-:|
> | PGD+EOT | 100 | 53.12 |
> | DiffAttack | 95.31 | 93.75 |
> | Targeted attack |95.31|95.31 |
> |C&W| 90.17±4.31 | 85.09±6.37 |
> |SPSA [R2-11]| 89.32±4.85 | 89.19±4.81 |
>
> [R1-1] Robust Evaluation of Diffusion-Based Adversarial Purification, ICCV 2023. (Lee & Kim (2023) in our paper) \
> [R2-6] DiffAttack: Evasion Attacks Against Diffusion-Based Adversarial Purification, NeurIPS 2023. (Kang et al. (2024) in our paper) \
> [R2-9] https://github.com/kangmintong/DiffAttack/blob/main/DiffAttack_Score_Based/diffattack/diffattack_base.py#L213 \
> [R2-10] Evaluating the adversarial robustness of adaptive test-time defenses, ICML 2022. (Croce et al. (2022) in our paper) \
> [R2-11] Adversarial risk and the dangers of evaluating against weak attacks, ICML 2018.

---

### Official Review · Reviewer_wQgS · 2024-11-01

**Soundness:** 3
**Presentation:** 2
**Contribution:** 2
**Rating:** 6
**Confidence:** 3

**Summary:**

This paper introduces a test-time adversarial defense method that utilizes diffusion-based purification along Opposite Adversarial Paths to excessively denoise adversarial inputs, pushing them away from decision boundaries. The proposed plug-and-play purifier can be integrated into pre-trained models, improving their robustness against adversarial attacks.

**Strengths:**

1. The proposed plug-and-play purifier can be integrated into pre-trained models, improving their robustness against adversarial attacks.
2. The method increases the time required for attackers to generate adaptive adversarial examples.
3. The paper also critiques the use of AutoAttack (Rand) for evaluating diffusion-based defenses and highlights the trade-off between attack effectiveness and computational complexity in adversarial robustness evaluations.

**Weaknesses:**

1.  The author's explanation of his method is not clear enough. There is no overall algorithm description (e.g. for section 3.3), making the reader difficult to follow.
2.  The explanation in Table 2 is not persuasive. Why with the increase of K, the robustness decreases?   It seems that this is contrary to the assumption in Figure 1.  Could you provide a more detailed explanation of this apparent contradiction and discuss potential reasons for the decrease in robustness as K increases?
3.  The algorithm is too complex to be applicable. For example, it uses several diffusion paths. From Table 6, it seems that the purification time will be doubled.  Could you discuss the trade-offs between complexity and performance? Are there ways to simplify the algorithm while maintaining its effectiveness? Additionally, it would be helpful to provide a more detailed analysis of the computational costs and how they scale with the number of diffusion paths.
4. The improvements are little compared with DiffPure regards the transferability. Moreover, this paper did not compare with the latest work, such as
[1] Robust Evaluation of Diffusion-Based Adversarial Purification,ICCV2023
[2] Adversarial Purification with the Manifold Hypothesis,AAAI2024
5.  References should be enclosed in parentheses. For example, it should be In DiffPure (Nie et al. 2022) rather than In DiffPure Nie et al. (2022).
6. Some typos. For example, in Figure 2, it should be Sec 3.xx.

**Questions:**

1. What is the aim of Color OT?
2. Why does the robustness decrease with the increase of K in Table 2?
3. How does the number of diffusion paths influence the purification time?

---

> ### Author Response · Authors · 2024-11-20
>
> **Response to W1 (Lack of algorithm in Sec. 3.3):** We have provided the pseudo-code of describing the proposed algorithm in Sec. 3.3 as Algorithm 1 in Sec. 12 of Supplementary (see revised PDF).
>
> **Response to W2/Q2 (Table 2):** Please also refer to our response to W1 of Reviewer xpXC.
> We respond from three perspectives.
>
> * First, we draw reviewers’ attention to Table 1, where the experiments were conducted on the test dataset under the following conditions:
>     1. We first attack the input with the adversarial attack and then move the attacked input with $-K$ steps while the label $y$ is known at every step.
>     2. The direction computed at each step is a gradient direction toward a lower loss area.
>     3. The data moves from $x_{adv} \to x \to x^1 \to \cdots \to x^K$ in a "step by step" manner.
>
>     This experiment aims to explore how the opposite gradient direction influences robust accuracy and, more importantly, to see if a safe zone exists for each data point. To utilize such gradient information surrounding the decision boundary, we proposed to learn the mapping $x_{adv} \to x^K$ by an NN model $g_\theta(\cdot)$. In other words, we want to use $g_\theta(\cdot)$ to approximate $K$ gradient steps.
> * On the other hand, the experiment in Table 2 was also conducted on the test dataset but with a learned $g_\theta(\cdot)$ under the conditions different from those in Table 1:
>     1. We first attack the input with the adversarial attack.
>     2. $g_\theta(\cdot)$ learns from $x_{adv} \to x^K$ only in one step, i.e., $x^K=g_\theta(x_{adv})$.
>     3. In testing stage, label $y$ is NOT given while we move the input with $g_\theta(\cdot)$.
>
>     Since label $y$ is unknown during inference, the goal of training $g_\theta$ is to fit the path from adversarial examples to an excessively denoised data via $K$-step. From the perspective of gradient descent, the step size is crucial to finding a better point for the next optimization step. When the step size is small, the start and end points approximate the linearity of the objective function. Connecting these small steps will form a curve or a zigzag path.  Such paths are non-linear but they describe complicated gradient information.
>
>   Yet, if one chooses a large step size, the algorithm might overshoot the better endpoint since a large step size ignores such detailed information around the endpoint (e.g. a zigzag path). In our explanation for Table 2 in the paper, we analogize such $x_{adv} \to x^K$ as a large step size in gradient descent when $K>1$. Therefore, it is too coarse for $g_\theta(\cdot)$ to capture the details of gradient information around the decision boundary if we use $x^K$, where $K>1$, as the endpoint for training a $g_\theta(\cdot)$. Hence, the large step effect and the lack of label guidance lead to decreased performance in Table 2 while $K$ increases.
> * In Table 1, due to the presence of $y$ and step-by-step update, we can find useful gradient information in every step. Therefore, using a purifier that maps $x_{adv} \to x^1$ (i.e. with a small step) and repeatedly applying such an update during the inference is a better way to capture gradient information. That is why we adopt this strategy alongside the reverse diffusion process of score-based diffusion for moving the adversarial data to a safer area.
>
> **Response to W3/Q3 (Complexity):** We respond to this comment in three aspects.
> * Please refer to Appendix 10 and Lines 881 $\sim$ 884, when the number of paths doubles, the computation complexity will also be doubled. This is due to the number of model parameters $N_{dm}$ is doubled too. Imagine a practical scenario that the attacker would like to generate a stronger attack, such as attacks with EOTs, by attacking our model. The input needs to be sent to our model multiple times so as to compute the corresponding gradients and then average the gradients with the purpose of producing a more effective adversarial example.
>
>   Obviously, such a procedure will consume more computation budgets for the attacks to pay than the inference time of our model (i.e., the ratio of attack-to-defense cost in Appendix 10). In other words, the computational burdens fall to attackers if they want stronger adversarial examples. Otherwise, they can use weaker attacks by reducing iterations or the number of EOTs. Hence, our method's goal is to deter attackers while maintaining effectiveness in terms of robust accuracy, as shown in our experiments.
> * We do not consider to simplify the algorithm since this will reduce attackers' burden, violating our goal of deterring attackers. For example, while we tested the AutoAttack(rand) on CIFAR-10 against DiffPure, it only took two days to finish. However, it takes more than a week to run such a test on our double-path diffusion (Sec 3.3).
> * Most works only evaluate the defense's effectiveness, yet our work provided an effective defense strategy and made a practical perspective to look at attackers' computation complexity.

---

> ### Author Response · Authors · 2024-11-20
>
> **Response to W4 (Comparison with the latest work):** We respond to this comment in three aspects.
> * First, the PGD-EOT attack in Table 5 of our submission is actually from [R1-1] (it was [1] mentioned in the reviewer's comment). Moreover, combining such an attack with our discovery and implementation, as described in Sec. 3.4, can yield a more exact attack gradient of an example, thereby creating stronger attack examples than those produced in [R1-1]. Please refer to Fig. 4 for the illustration of gradient computation of different strategies.
> * Second, regarding the transferability, we guess the reviewer refers to Table 6 in [R1-1]. Under $t^*=100$, which we used in our experiments, the performance seems lower than ours. Moreover, higher robust accuracy in that table suggests that "Although the transferred attack is valid, the attack success rates using different samplers are slightly lower than those using the original samplers" by authors of [R1-1]. Hence, in our opinion, to not overestimate the robust accuracy, one should evaluate the defense with the reverse diffusion strategy consistent with the one use in that defense.
> * Third, there are two reasons that the evaluation of [R1-2] (it was [2] mentioned in the reviewer's comment) does not meet our requirements.
>   1. Classifiers' architecture allegedly used in [R1-2] is inconsistent with those used in other works. Most of the works used classifiers provided by [R1-4]. The comparison won't be fair for both parties in such a scenario.
>   2. The white-box adaptive attacks they used do not combat the randomness of the defense with Expectation over Transformation (EOT). According to [R1-3], one must include EOT to avoid overestimating the robustness of defenses with randomness.
>
>   Even though the above inadequacy is not considered, their performance in terms of robust accuracy against standard attacks and adaptive attacks does not compete with ours (Table 2 in [R1-2]: $\bf{60.52}$ vs. Table 5 in ours: $\bf{81.25\pm3.62}$).
>
> **Response to W5/W6:** We have revised the typos and corrected the writing. Thanks for pointing out them for further improving the writing quality of this paper! (see revised PDF)
>
> **Response to Q1 (Aim of color OT):** Since we don’t have clean data or a better starting point other than the adversarial example during testing, we use color OT to wash the adversarial example and use it as the surrogate starting point.
>
> **Refs:** \
> [R1-1] Robust Evaluation of Diffusion-Based Adversarial Purification, ICCV 2023. (Lee & Kim (2023) in our paper)\
> [R1-2] Adversarial Purification with the Manifold Hypothesis, AAAI 2024. \
> [R1-3] Obfuscated Gradients Give a False Sense of Security: Circumventing Defenses to Adversarial Examples, ICML 2018. (Athalye et al. (2018) in our paper) \
> [R1-4] https://robustbench.github.io/ (Croce et al. (2021) in our paper)

---

> > ### Comment · Reviewer_wQgS · 2024-11-22
> >
> > Thanks for your response. Most of my concerns have been resolved, so I have changed my score to 6.

---

> > > ### Author Response · Authors · 2024-11-24
> > >
> > > Dear Reviewer wQgS,
> > > Thank you so much for your active and positive feedback! It encourages us a lot.
> > >
> > > Best,
> > > Authors

---

### Author Response · Authors · 2024-11-21

Dear SAC/AC/Reviewers,
The authors would like to thank the reviewers for their valuable comments that help us to further clarify/explain the contributions of our submission.
The authors appreciate:
(A)	Reviewer wQgS for recognizing that our method is able to (1) improve robustness against adversarial attacks, (2) increase the time for the attackers to create the adaptive adversarial attacks, and (3) highlight the tradeoff between attack effectiveness and computation complexity;
(B)	Reviewer vNum for recognizing (1) our idea of reducing adversarial perturbations, and (2) the effectiveness of our method; and
(C)	Reviewer xpXC for recognizing that our work (1) introduces a new method to alleviate the influence of adversarial noises, (2) provides extensive evaluation results, and (3) presents multiple related works for comparison.

For the major concerns, regarding (1) the possible contradictive statements in Sec. 3.1.2 and Sec. 1.3, (2) reducing the complexity of proposed method, and (3) evaluation on more/different adversarial attacks, we will clarify and explain in the following in details.

Sincerely,
Authors

---

> ### Author Response · Authors · 2024-11-24
>
> Dear Reviewer xpXC and Reviewer vNum,
>
> Thank you very much for your valuable comments on our submission!
> We have clarified and responded to your questions, and would like to know if our responses satisfy you.
> We are happy to reply to any unclear parts, if any, since the last day for reviewers to ask questions to authors is approaching!
>
> Best,
> Authors

---

> > ### Author Response · Authors · 2024-11-30
> > **New experimental results**
> >
> > Dear Reviewer xpXC and Reviewer vNum,
> >
> > The authors spent a lot of time to address your concerns by conducting new experiments.
> > The results are promising, and validate the effectiveness of our method.
> >
> > Could you please take a look at them?
> >
> > Since the deadline of interaction between the reviewers and authors is approaching, we appreciate your time and effort!
> >
> > Sincerely,
> > Authors of 10376

---

### Meta-Review · Area_Chair_CEov · 2024-12-20

**Metareview:**

This work proposed a test-time adversarial defense method via diffusion-based recovery along opposite adversarial paths.

It received 3 detailed reviews. The ideas and the effectiveness in the reported experiments are recognized by most reviewers. Meanwhile, there are also several important concerns, and I think the most important two are:
1. Threat model: The method is only evaluated on white-box gradient-based non-targeted attacks.
2. Complexity: The complexity is hight.

The authors made efforts to provide rebuttals, and all reviewers gave further feedback or updated their reviews. Combing reviewers' feedback and my judgement, the final comments about above two points are:
1. It is very strange that the test-time adversarial defense is evaluated under the white-box setting, which means that both attacker and defender can access the model during the whole process, like an online gaming. I cannot imagine when it could happen in real scenarios. The practical setting should be test-time adversarial defense against black-box attacks, and both attacker and defender don't know each other's existence and strategy. Based on the impractical setting, the evaluations of this work are unconvincing.
2. The authors argued in the rebuttal that the additional complexity for defense is not important. It makes no sense. A good defense should neither increase too much cost nor harm the model's performance on benign query.

Based on above two points, I think the practical effectiveness of the proposed method is limited. Thus, the recommendation is reject.

**Additional Comments On Reviewer Discussion:**

The rebuttal and discussions, as well as their influences in the decision, have been summarized in the above metareview.

---

### Decision · Program_Chairs · 2025-01-22

Reject